# Application of Natural Language Processing (NLP) in Detecting and Preventing Suicide Ideation: A Systematic Review

**DOI:** 10.3390/ijerph20021514

**Published:** 2023-01-13

**Authors:** Abayomi Arowosegbe, Tope Oyelade

**Affiliations:** 1Institute of Health Informatics, University College London, London NW1 2DA, UK; 2Division of Informatics, Imaging & Data Sciences, University of Manchester, Manchester M13 9PL, UK; 3Division of Medicine, University College London, London NW3 2PF, UK

**Keywords:** natural language processing, NLP, text mining, suicide prevention, suicide-ideation, mental health

## Abstract

(1) Introduction: Around a million people are reported to die by suicide every year, and due to the stigma associated with the nature of the death, this figure is usually assumed to be an underestimate. Machine learning and artificial intelligence such as natural language processing has the potential to become a major technique for the detection, diagnosis, and treatment of people. (2) Methods: PubMed, EMBASE, MEDLINE, PsycInfo, and Global Health databases were searched for studies that reported use of NLP for suicide ideation or self-harm. (3) Result: The preliminary search of 5 databases generated 387 results. Removal of duplicates resulted in 158 potentially suitable studies. Twenty papers were finally included in this review. (4) Discussion: Studies show that combining structured and unstructured data in NLP data modelling yielded more accurate results than utilizing either alone. Additionally, to reduce suicides, people with mental problems must be continuously and passively monitored. (5) Conclusions: The use of AI&ML opens new avenues for considerably guiding risk prediction and advancing suicide prevention frameworks. The review’s analysis of the included research revealed that the use of NLP may result in low-cost and effective alternatives to existing resource-intensive methods of suicide prevention.

## 1. Introduction

Suicide is the world’s 13th leading cause of death, accounting for 5–6 percent of all fatalities [1]. The likelihood of completing suicide varies by sociodemographic variables around the world, with young adults, teenagers, and males bearing the largest risks [2]. Every suicide is a tragedy that impacts families, towns, and whole nations, as well as the individuals who are left behind by the deceased. Suicide occurs at any age and was the fourth highest cause of death among 15–29 years old worldwide in 2019 [3]. Because of the COVID-19 pandemic, people all over the world have been suffering from the effects of the financial crisis, mental health issues, and a sense of loneliness and isolation. These factors have heightened public awareness of the dangers of suicide. Suicidal behaviour is complicated, and no one explanation fits every case. However, many people commit suicide on the spur of the moment, and having ready access to a means of suicide, such as poisons or weapons, may make the difference between life and death [4]. Attempting suicide by other ways, such as jumping in front of a speeding train or plunging from tall buildings, has also been reported [4]. Thus, removing the means of suicide may not significantly reduce the rate of suicide.

Suicide is a severe public health issue, but it is avoidable with early, evidence-based, and frequently low-cost measures. A robust multi-sectorial suicide prevention plan is required for national suicide interventions to be successful [3]. Innovative and cost-effective ways to collect and understand data for suicide prevention are important tools in the fight against suicide [5]. Approaches such as NLP combined with other machine learning techniques that utilise existing data from Electronic Medical Records (EMRs) and other repositories have the capability to improve early identification of people at higher risk of committing suicide. This is especially true given that these computational approaches can provide a low-cost alternative to other costly methods [6]. Text mining approaches that are now in use include information retrieval, text classification, document summarisation, text clustering, and topic modelling. These approaches, on the other hand, concentrate on collecting usable information from text documents using a range of techniques such as keyword extraction, categorisation, topic modelling, and sentiment analysis [7]. These approaches, in contrast to NLP, are more limited in scope and do not always focus on comprehending the meaning of texts. NLP, on the other hand, can comprehend the meaning and context of words, as well as the mood and emotion behind texts, phrases, and sentences. This enables it to understand complicated texts more effectively and extract more relevant insights than conventional text mining approaches [8].

Over the past several decades, there has been a significant expansion in the body of knowledge about suicidal behaviour. For instance, research has revealed that the interaction of biological, psychological, social, environmental, and cultural elements is an important component in influencing suicide ideation [9]. At the same time, the field of epidemiology has been instrumental in determining a wide variety of variables, both protective and risky, that influence the likelihood of an individual committing suicide, both in the general population and in specific susceptible groups [5,10]. It has also come to light that the risk of suicide varies greatly among cultures, with culture playing a role both in elevating the risk of suicidal behaviour and in providing some protection against it [10].

In terms of legislation, it is now known that 28 countries have national suicide prevention policies, and World Suicide Prevention Day, which is celebrated annually on September 10 and is coordinated by the International Association for Suicide Prevention, is recognised all over the world. In addition, a great number of research centres devoted to suicide have been established, and there are academic programmes that concentrate on the prevention of suicide [4]. Self-help groups for the bereaved have been created in several different locations, and trained volunteers are assisting with online and telephone counselling services to provide practical assistance. Non-specialized health professionals are being used to strengthen the evaluation and management of suicidal behaviours. Decriminalizing suicide in many countries over the course of the last half-century has made it considerably simpler for those who struggle with suicidal tendencies to get the assistance they need [4].

For suicide prevention strategies to be successful, there must be an improvement in surveillance and monitoring of suicide and attempts at suicide. Healthcare providers and treatment facilities need access to innovative tools that will help persons who are at risk of committing suicide get mental health care and continue to be safe until they do [3]. According to the National Institute of Health NIH, there are two primary methods for identifying who is at risk of committing suicide: first, “Universal Screening”, which, according to some estimates, has the potential to identify more than three million adults who are at risk of committing suicide annually. The second primary method for identifying who is at risk of committing suicide is by “Predicting Suicide Risk using Electronic Health Records”. The use of electronic medical records, including the unstructured text of patients’ medical notes such as discharge summaries, is recognised as a vital resource for the provision of medical treatment as well as for medical research [11].

The extraction of information and the discovery of new knowledge using NLP and other machine learning methods have been successfully applied to electronic medical notes and other text data in a variety of mental health areas such as depression [12] and post-traumatic stress disorder (PTSD) [13]. An NLP model that recognises indicators of sadness in free text, such as posts in internet forums like twitter and reddit, chat rooms, and other such sites, has been developed. Machine learning and artificial intelligence approaches were used to create this model. NLP was also used to extract emotional content from textual material to identify patients with PTSD using sentiment analysis from semi-structured interviews; a machine learning (ML) model was trained on text data from the Audio/Visual Emotion Challenge and Workshop (AVEC-19) corpus [14].

Suicides can be prevented, and there have been several measures and screening methods that have been used in the past [4,11]. These include limiting access to the means of suicide (such as pesticides, weapons, and certain medicines), training and education of healthcare professionals in recognising suicidal behaviour, responsible media reporting, raising awareness, and the use of mobile apps and online counselling tools, amongst other potential solutions. However, the screening tools that are now available may not be sensitive enough to enable person-centred risk detection consistently [15]. Consequently, there is an urgent need for novel approaches that focus on the individual when identifying people who may be at risk for suicide. To improve upon how things are done and to have an impact on policy, the purpose of this project is to search for, analyse, and report on ways suicide may be prevented using NLP.

### 1.1. Rationale

As this is a pressing challenge in the United Kingdom and around the world, it is necessary that more research and studies be carried out to slow down the growing number of individuals who take their own lives.

It is difficult to detect suicide ideation because people who are suicidal tend to isolate themselves and are unwilling to communicate about their thoughts [16]. As a result, detecting suicide ideation may be extremely challenging. Those who are at risk of committing suicide need to be monitored constantly to identify when they are having suicidal thoughts so that appropriate action may be taken. This may allow healthcare professional and relevant experts to save lives through timely interventions.

According to the National Institutes of Health NIH, utilising electronic medical records is one of the ways that suicide might be averted [11]. However, there hasn’t been enough of work done in this area, especially using text analytics tools like NLP. The development of a risk stratification tool via the use of electronic medical records, including both structured and unstructured data, is one method that may be used to reduce the incidence of suicide. To contribute to the growing research landscape that could aid in the development of a suicide prevention tool, the purpose of this study is to investigate and consolidate essential work that has been done on the use of NLP for detecting suicidal thoughts.

### 1.2. Research Question

Can natural language processing (NLP) and other text analytics methods be used in identifying people with suicide ideation?

### 1.3. Aims & Objectives

Machine learning and artificial intelligence-based modelling, such as NLP and other text analytics approaches, have the potential to become major techniques for the detection, diagnosis, and treatment of people who are suffering from mental health issues [17]. This was demonstrated by the results of research in mental health such as depression, post-traumatic stress disorder (PTSD), and homelessness [18,19,20]. The primary aims of this research is to assess how NLP has been utilised in the field of suicide prevention and its effectiveness as well as limitations. The goal is to provide recommendations for improvement and suggest areas needing further research. The objectives of this research are listed below.

Conduct a comprehensive database search for research on the use of NLP for suicidal ideation.Collect essential information on the detection and treatment effectiveness of the NLP approach, as well as its limitations.Synthesize, analyse, and report findings from included studies.Make future suggestions and identify prospective research areas.Formulate recommendations for future efforts based on the findings of the included studies.

## 2. Methodology

This was a qualitative study with the goal of completing a review of studies that had been conducted using NLP and other text analytics approaches in the identification or detection of suicidal ideation. This systematic review was carried out in accordance with the PRISMA (Preferred Reporting Items for Systematic Reviews and Meta-analyses) standards to increase both the level of transparency and the quality of the reporting on publications [21].

### 2.1. Inclusion & Exclusion

All peer-reviewed journal publications published during the last 10 years were included. Also included are articles written or translated for publication in the English language. In addition, studies addressing the use and application of NLP methods or other text mining approaches for suicide, suicide ideation, and self-harm in any environment, including mental health, acute, and community services, were included.

We excluded research that did not include NLP or other text mining techniques. In addition, reviews and other secondary sources were excluded. All poster presentations, non-full-text submissions, and full-text submissions in languages other than English were also excluded.

### 2.2. Search Strategy

Using the following syntax, the Population, Phenomena of Interest, and Context (PICo) framework was used for the search approach.

((NLP OR “natural language processing” OR “text mining” OR “text analytics” OR “data mining” OR “information retrieval”)

AND

(“mental health” OR disorder OR depression OR suicide OR psychotic OR psychiatry OR “self-harm” OR suicidal))

### 2.3. Databases

A search of the relevant literature was conducted for this investigation utilising five scientific and medical databases: PubMed, MEDLINE, Embase, PsycINFO, and Global Health through the OVID platform. Several papers discovered in the reference lists of the included studies were also included.

### 2.4. Reference Management

Mendeley was used for reference management, paper collection, and organisation. Mendeley was chosen for this study because it allows researchers to import and store papers from a variety of databases and in a variety of formats. It can also be used to remove duplicates, especially in the case of papers that appear in multiple databases, and export papers to other applications, such as systematic review management systems such as Covidence and Rayyan.

Covidence was utilised to manage the systematic review. It’s a web-based systematic review management software that makes it easier to create systematic reviews and other types of research reviews by screening citations and completing texts, assessing bias risk, and extracting study features and findings for analysis. The Covidence system was chosen because it speeds up the initial screening of abstracts and full texts, enabling the author and the second reviewer to collaborate on the project and resolve any disagreements about whether papers should be included or excluded.

Using Covidence, two independent researchers (AA and TO) reviewed the publications at the abstract and full-text stages in line with the inclusion criteria. Conflicts were resolved at each stage of the review until consensus was obtained. Cohen’s Kappa Coefficient, which measures the degree to which the data gathered reflects the variables tested, was used to examine interrater reliability.

### 2.5. Quality Assessment

The quality of the included publications in this research was evaluated using the Mixed Method Appraisal Tool MMAT. The mixed-method evaluation instrument is used to evaluate quantitative, qualitative, and mixed-method studies that are included in systematic reviews of mixed-studies [22]. With a focus on mixed-methods research, the tool specifies a series of criteria and screening questions to obtain an overall quality score. The MMAT evaluation tool was used since it is an appropriate evaluation instrument for systematic reviews that include various study designs such as qualitative research, randomised controlled trials, nonrandomized studies, quantitative descriptive studies, and mixed methodologies research etc.

### 2.6. Databases

To obtain relevant data from the included studies, a template for data extraction was designed. After being exported from Covidence software, the data were cleaned and transformed in Microsoft Word and Excel before analysis. After the data had been exported into the data processing software, it was examined and investigated to determine whether the appropriate data had been collected. A matrix was then created to store the data, initial codes were derived from the data, the codes were examined, revised, and combined into themes, and, finally, the themes were refined and presented in a cohesive manner.

### 2.7. Analysis

Thematic analysis using the reflective approach was used to construct narratives and discussions from the included papers, using codes and themes generated from the collected data. These narratives and discussions were based on the findings of the included studies. For the purposes of this research, specialised software was not required to carry out the thematic analysis. Instead, tables were created in Microsoft Word, which serve as the repository for the core themes as well as the secondary themes.

Reflective thematic analysis RFA was adopted for this research because of its widespread use and reputation as one of the more accessible methods for those with little or no prior experience in qualitative analysis [17]. In addition, reflective thematic analysis allows easy identification and analysis of patterns or themes in a given data set and also provides a simple and theoretically flexible interpretation of qualitative data [18,19].

### 2.8. Ethics

There are no ethical issues about the safety of the participants or the data collected in this research. Full-text literature was obtained from several medical, health informatics, and psychological sources available via the university library and other third-party databases. Therefore, the data and information gathered by this study are already accessible in the public and academic domains. The lead investigators of the included studies are expected to have obtained consent from all persons, organisations, and subjects participating in their investigations. As a result, no ethical approval is required for this systematic review.

## 3. Results

### 3.1. Study Selection

The preliminary search, which consisted of searching 5 separate databases with the help of the OVID platform, produced a total of 387 results. After the processing of the information in Mendeley reference management software, a total of 158 records were produced after the deduplication and initial screening process.

The 158 data that had been pre-processed in Mendeley were then imported into Covidence, which is a management system for systematic reviews, and here is where the screening processing was completed. Following the review of the full text, twenty (20) studies were assessed and chosen for inclusion. The search procedure is shown in Figure 1 using the PRISMA flow diagram shown below.

### 3.2. Study Characterisitics

The characteristics of the included (*n* = 20) studies are outlined in Table 1. Most studies (*n* = 12, 60%) were conducted in the United States, although four studies were conducted in the United Kingdom, two in Asia, one in Spain, and one in Brazil. A total of 50% of the included studies (*n* = 10) were done in a clinical context. A total of 15% of the studies were conducted online or utilising mobile apps. Two studies were conducted in an emergency department setting. Studies including the modelling of EHR data (*n* = 8), qualitative interviews (*n* = 2), and 10 experimental studies (*n* = 10).

The Figure 2 below depicts the included studies by setting, with research done in a clinical context being the largest proportion (*n* = 10).

The Figure 3 below depicts the breakdown of studies by country in which they were conducted, with the United States having the most (*n* = 12).

### 3.3. Screening in Emergency Departments

The unexpected nature of suicide makes it a leading cause of death, which complicates efforts being made all over the world to prevent it [37]. In recent years, the ability to analyse large datasets using machine learning and artificial intelligence (ML/AI) has been possible, which results in improved risk detection. Patients who attempt suicide may seek help from nearest emergency department, and their chances of survival are dependent on successful assessment and treatment. Indeed, most completed suicides are results of repeated attempts made by undetected and untreated individuals [38,39]. Estimating the likelihood of multiple suicide attempts is largely left to clinical judgement in the Emergency Department, where suicidal patients often appear [38]. Thus, early recognition of self-harm presentations to emergency departments (ED) may result in more prompt suicide ideation care.

The research investigated whether NLP/ML used on recorded interviews for suicide risk prediction model can be implemented in two emergency departments in the South-eastern United States. In the research, interviews were conducted with 37 suicidal and 33 non-suicidal patients from two emergency departments to evaluate the NLP/ML suicide risk prediction model [28,40]. The area under the receiver operating characteristic curve (AUC) and Brier scores were used to assess the model’s performance. The research demonstrates that it is viable to integrate technology and methods to gather linguistic data for a suicide risk prediction model into the emergency department workflow. In addition, a fast interview with patients may be used efficiently in the emergency department, and NLP/ML models can reliably predict the patient’s suicide risk based on their comments.

Similar to [25], [28] performed a prospective clinical trial to examine whether machine learning techniques may distinguish between suicidal and non-suicidal people by based on their conversations. NLP and semi supervised machine learning techniques were used to record and evaluate the discussions of 30 suicidal teenagers and 30 matched controls using questionnaires and interviews as the data collection tools. The findings demonstrates that the NLP model successfully differentiated between suicidal and non-suicidal teenagers.

### 3.4. Avoiding Perinatal Suicide

Neonatal fatalities decreased worldwide by 51%, from 5 million in 1990 to 2.5 million in 2017 [41]. However, this drop has not been seen in low-income and middle-income nations, which have the largest burden of neonatal deaths [42]. It has been shown that prompt delivery of high-quality healthcare services and early identification of pregnant women at risk for unfavourable maternal and perinatal outcomes throughout the prenatal period enhance mother and neonatal survival [43]. In low-resource countries where the bulk of perinatal fatalities take place at home, machine learning and artificial intelligence models may be a crucial tool in assessing risk factors for perinatal mortality and triaging pregnant women at high risk of severe postpartum depression, suicide ideation, and death.

An NLP tool developed by [29] was used to detect perinatal self-harm in electronic health records with a sufficient level of recall and accuracy. Additionally, depending on their EHR, authors identified service users who have self-harmed during pregnancy using the NLP tool. The work demonstrates that it is possible to create an NLP tool that can recognise instances of perinatal self-harm in EHRs with acceptable validity; however, there are certain temporal restrictions.

Further, Zhong et al. [34], successfully created algorithms that used data from clinical notes collected using NLP in electronic medical records to detect pregnant women who were exhibiting suicidal behaviour. They used both structured, codified data and unstructured, NLP-processed clinical notes to extract diagnostic information for their investigation, and they evaluated the algorithm’s diagnostic validity in comparison to gold-standard labels generated from manual chart checks by psychiatrists and a skilled researcher. The study also demonstrated that using structured data and employing NLP to mine unstructured clinical notes significantly enhances the ability to identify suicidal behaviour in pregnant women. In addition, the approach led to an 11-fold increase in the number of pregnant women whose suicide behaviours were identified.

Similarly, [30] compared the performance of predefined diagnostic codes vs. NLP of unstructured text for detecting suicidal behaviour in pregnant women’s electronic health data. Utilizing NLP significantly increases the sensitivity of screening for suicidal behaviour in EHRs. Nevertheless, the proportion of verified suicidal behaviour was lower among women who did not have diagnostic codes for suicidal behaviour.

### 3.5. Digital Applications for Suicide Detection

Regrettably, only a small percentage of suicidal patients actively participate in their therapy, and this percentage is much lower for patients whose suicidal ideation is both frequent and strong. However, although some individuals with a high suicide risk avoid face-to-face intervention, they may be more likely to try to get aid discreetly via technological means [44].

Mobile health applications (MHA) have the potential to expand access to evidence-based care for those who have suicidal thoughts by addressing some of the constraints that are present in traditional mental health therapy [45]. These obstacles include stigmatisation, the perception that expert treatment is not required, and inadequate time in an acute suicidal crisis. The proliferation of smartphones has made MHA possible. As a result, the MHA can deliver assistance in a timely manner, in a convenient manner, in a discrete manner, and at a cheap cost, particularly in a severe crisis, since they are not constrained by time or location [46].

In the study by [26], an automated algorithm for analysing and estimating the risk of suicide based on social media data was developed. The research investigates how the technique may be used to enhance current suicide risk assessment within the health care system. It also explores the ethical and privacy considerations associated with developing a system for screening undiagnosed individuals for suicide risk.

The research indicates that the technology can be used for intervention with people who have decided not to opt in for interventional services. Indeed, technology allows scalable screening for suicide risk, with the possibility to identify many people who are at risk prior to their engagement with a health care system. However, although the development of the intervention system based on algorithmic screening is technologically possible, the cultural ramifications of its implementation are not yet decided.

Further, [15] developed the Boamente program, which gathers textual data from users’ smartphones and detects the presence of suicidal ideation. They created an Android virtual keyboard that can passively gather user messages and transfer them to a web service using NLP and Deep Learning. They then created a web platform that included a service for receiving text from keyboard apps, a component with the deep learning model implemented, and a data visualisation application. The technology exhibited the capacity to detect suicidal thoughts from user messages, nonclinical texts, and data from third-party social media apps such as Twitter, allowing it to be tested in trials with professionals and their patients.

Like [15], [5] employed NLP and machine learning to predict suicide ideation and elevated mental symptoms among adults recently released from psychiatric inpatient or emergency hospital settings in Spain. They used NLP and ML (logistic regression) on participant-sent text messages. The text message included a link to a questionnaire and a mobile application for collecting participant replies. The research demonstrates that it is feasible to apply NLP-based machine learning predictions algorithms to predict suicide risk and elevated mental symptoms from free-text mobile phone answers.

A domain Knowledge Aware Risk Assessment (KARA) model is created in experimental research by [35] to enhance suicide identification in online counselling systems. In their research, they used NLP on a de-identified dataset of 5682 Cantonese talks between help-seekers and counsellors from a Hong Kong emotional support system. The study show that it is both feasible and beneficial to utilise an accurate, passive, and automated suicide risk detection model to inform counsellors of potential risks in a user’s information as they are engaging with the user. Additionally, the NLP model performed better than traditional NLP models in several experiments, indicating strong clinical relevance and translational utility.

### 3.6. Suicide Prevention Using Electronic Health Records EHR

Electronic health record (EHR) data, in addition to a clinical decision support system (CDSS), may act as an “early warning system” to notify professionals about patients who should be evaluated for suicide risk [47,48]. CDSS is a health information system that may be incorporated into EHR system or healthcare workflow, allowing clinicians to utilise it easily and effectively. Because of its capacity to deliver evidence-based healthcare to the point of treatment, the usage of these technologies has increased in recent years [49].

In four of the included studies [9,16,23,24], NLP was used on clinical notes obtained from electronic health records (EHR), such as the Clinical Record Interactive Search (CRIS) system, to identify patients who are at risk of suicidal ideation. Using NLP approaches, these investigations demonstrated the potential application of EHR information to further research on suicidality and self-harm. Accordingly, this technology also has the potential to be useful in the expansion of risk prediction in several other areas of mental health such as eating disorders and depression.

In addition to utilising clinical notes extracted from EHR, McCoy and colleagues [31] used sociodemographic data, billing codes, and narrative hospital discharge notes for each patient taken from the electronic health records (EHRs) of the hospital in order to enhance suicide risk prediction. The research demonstrates that utilising textual data other than clinical notes, such as demographic, diagnostic code, and billing data, might help clinicians in assessing suicide risks and may help in identifying high-risk people with high precision. Using psychotherapy and psychiatric data from EHRs might also potentially enhance suicide risk prediction, as shown by [19,36]. Indeed, [36] extracted EHR data of hospitalised patients and PTSD patients and applied NLP, SVM, KNN, CART, Logistic Regression, RF, Adaboost, and LASSO for a suicide risk prediction tool. The results imply utilising NLP and data from psychotherapy and psychiatric notes to automatically categorise patients with or without suicide ideation before hospitalisation, could possibly result in significant time and resource savings for the identification of high-risk patients and the prevention of suicide.

### 3.7. Racial Disparity

Suicide prevention initiatives must be more carefully targeted if they are to be successful. Racial and ethnic disparities in rates of suicidal thoughts, suicide attempts, and suicide fatalities need to be better understood [50]. The likelihood of suicide varies across racial and ethnic groups depending on their experiences with prejudice, past trauma, and the availability of culturally appropriate mental health care. At the same time, under reporting and other shortcomings in data collection methods restrict the understanding of racial and ethnic disparities in suicide and suicidal behaviours [51,52].

For instance, studies have shown that young people and women are more likely to engage in suicide behaviours [53,54]. The connection between race, ethnicity, and suicide ideation, on the other hand, is far less well understood. Researchers [10] undertook a study to enhance the accuracy of the categorization of undetermined-intent fatalities and to evaluate racial disparities in misclassification. National Violent Death Reporting System (NVDRS) restricted-access case narratives of suicides, murders, and undetermined deaths in 37 states were subjected to NLP and statistical text analysis. Their analysis demonstrates that the identification of suicide in deaths involving Black decedents can be significantly improved by employing race-specific death narratives in modelling. In their experiment employing race-specific narratives, the suicide prediction rate was significantly improved, and prediction power was found to be comparable to that for unexplained deaths involving white decedents. Overall, there is strong evidence that NLP and automated coding systems might improve the detection of suicide warning signals. Also, including race as a variable in the modelling method could result in improved predictive power and consequently a reduction in suicide ideation and deaths, especially among groups at higher risk of suicide.

### 3.8. Quality Assessment

After an MMAT evaluation, the studies that were included were found to be satisfactory. On the other hand, some studies included a very limited description of the methodology, population, and settings in which their study was carried out.

## 4. Discussion

This is the first qualitative systematic review on the use of NLP for suicide prevention. Using both structured and unstructured data in data modelling with NLP yielded much more accurate results, as compared to using either structured or unstructured data alone. Multiple studies demonstrate that integrating structured data, such as diagnosis code, demographics, and billing data, with unstructured data, such as narratives, increases the performance and accuracy of detecting individuals with suicidal ideation.

Additionally, persistent and passive observation of individuals with a confirmed diagnosis of mental health issues is essential and is shown to reduce suicide and self-harm incidence [15,35]. It has been reported previously that up to ninety percent of suicides are associated with mental health issues [55]; therefore, passively monitoring persons with a confirmed diagnosis using an NLP or other ML/AI-based suicide risk assessment tool might be useful and advantageous. However, ethical and privacy problems should be examined regarding the use of patient data for suicide surveillance or monitoring, and additional research is necessary to impact government policy in this area.

In addition, EHRs of ethnic minority patients have been reported previously to include less notes and details than those of non-ethnic patients [51]. Data equality is vital for reducing and preventing suicidal ideation, maximising the potential of NLP and other machine learning and artificial intelligence technologies, addressing health disparities, and ensuring fair access to services. Further, the use of race-specific data in the development of suicide risk or prediction systems might boost accuracy and performance and simultaneously avoid racial bias, as shown by [10,15]. Researchers and software developers should be conscious of this information, which has been found to improve the effectiveness of predictive tools [10]. Electronic health records have a variety of information, which is crucial for building a suicide risk assessment tool. In addition to EHR, it is also feasible to utilise social media and smartphone applications data to detect individuals with suicide ideation. Sometimes, to escape the societal stigma associated with suicide thoughts, people may use online platforms such as blogs, tweets, and forums to express themselves [45,46]. Therefore, including smartphones and social data in NLP models may enhance suicide diagnosis.

This research validates the results of a previous systematic reviews by [56], which concluded that NLP can be used in detecting and treating mental health issues including suicide and self-harm. In addition, NLP techniques may provide insights from unexplored data such as those from social media and wearable devices that are often inaccessible to care providers and physicians. Indeed, although machine learning and artificial intelligence solutions are not intended to replace clinicians in the prevention of suicide or other mental health issues, they can be used as a supplement in all phases of mental health care, including diagnosis, prognosis, treatment efficacy, and monitoring.

Natural language processing (NLP) is a powerful text mining method that has many benefits over other text mining methods. One of the main advantages of NLP is its ability to process and understand natural language. NLP algorithms are designed to identify meaning and structure in unstructured text, which makes it easier for the algorithm to accurately categorise and classify the data [57]. Additionally, NLP can interpret the context of language and understand the nuances of human communication, including the use of slang, sarcasm, and context-specific expressions. This makes it far more effective for extracting meaningful insights from text than other text mining methods such as sentiment analysis, information extraction, and text classification, among others. This advantage is probably a reason for the wide use of NLP in clinical diagnosis, especially in mental health [58,59].

Despite the many benefits of NLP, it also has a few limitations when compared to other text mining methods. One limitation is that NLP is limited to the language it is designed for; it cannot process text written in a language other than the language it was designed for. Additionally, NLP requires a large amount of data to be effective and can be difficult to interpret due to its complexity. Finally, NLP algorithms are often computationally intensive, requiring a considerable amount of computing power to process the data [60]. However, NLP use in suicide ideation have shown continued progress and is likely to improve the future of diagnosis and prevention of suicide ideation and related death.

Suicide ideation is increasingly being detected in social media postings, text messages, and other digital sources using supervised text mining approaches [61]. Supervised approaches categorise data using predetermined labels and can be used to find patterns in text that may indicate suicidal intent. One advantage of supervised approaches is that they may be used to find patterns that are not always visible and to uncover subtle nuances in language that may signal suicidal ideation. Furthermore, because the specified labels provide a more solid basis for assessment, supervised methods are more accurate than unsupervised approaches [62].

Unsupervised text mining approaches, on the other hand, find patterns in text without using predefined labels. The advantage of unsupervised approaches is that they may be used to swiftly examine huge amounts of data and find subtle patterns that may not be visible to the human eye. However, because there is no reliable basis for evaluation, unsupervised approaches are less accurate than supervised methods and are more prone to false positives [63]. 

### Limitations

This study has some limitations. Firstly, preprint and unpublished paper were not included in this study. Thus, grey studies and other data may exist which are not covered herein. However, Ovid-based databases including MEDLINE and Embase were systematically searched, and retrieved studies were subjected to manual reference search. Secondly, the methodologies utilised in the included research are too heterogeneous, and there are no metrics available to assess their efficacy. As a result, meta-analysis could not be conducted. Lastly, the efficacy of NLP in preventing suicide ideation and self-harm could not be quantified, as the included studies did not provide any metrics to the effect. Future study should give this a high priority since it might provide additional information regarding the efficacy of NLP in mental health.

## 5. Conclusions

According to the findings of this research work, NLP could help in the early detection of individuals who have suicide ideation and allow timely implementation of preventive measures. It is also found that passive surveillance via mobile applications, online activity, and social media is feasible and may help in the early diagnosis and prevention of suicide in vulnerable groups. However, before passive surveillance can be clinically useful, ethical and security issues need to be addressed.

When modelling, employing race specific terminologies has been demonstrated to boost both performance and accuracy among ethnic minority groups. This may boost health equality and allow equitable access to healthcare services. Furthermore, combining structured and unstructured data have been reported to enhance accuracy and precision in suicide detection, which is important for developing an NLP model for predicting suicide risk.

In summary, the application of artificial intelligence and machine learning offers new prospects to significantly enhance risk prediction and suicide prevention frameworks. Based on included studies, the use of NLP may be used to develop low-cost, resource-efficient alternatives to conventional suicide prevention measures. Thus, there is significant evidence that NLP is beneficial for recognising individuals with suicidal ideation, consequently giving unique opportunities for suicide prevention.

### Recommendations

Based on the results that were obtained from this review, the following recommendations have been made:Reducing suicide is a collective effort; the government should form a suicide prevention task group under DHSC to explore technical solutions for early suicide detection.Since most people with suicide ideation seek help from ED first, integrating NLP-based CDSS in ED workflow for suicide risk might help identify them early.Adequate training should be giving to staff to recognise unconscious racial bias when using EHR systems to record patients’ data.Include race-specific data in EHR systems and utilise them as a standard for developing suicide risk prediction tools.More study is required to explore privacy issues and ethics of passive data surveillance or monitoring, particularly on those with mental illness.

## Figures and Tables

**Figure 1 ijerph-20-01514-f001:**
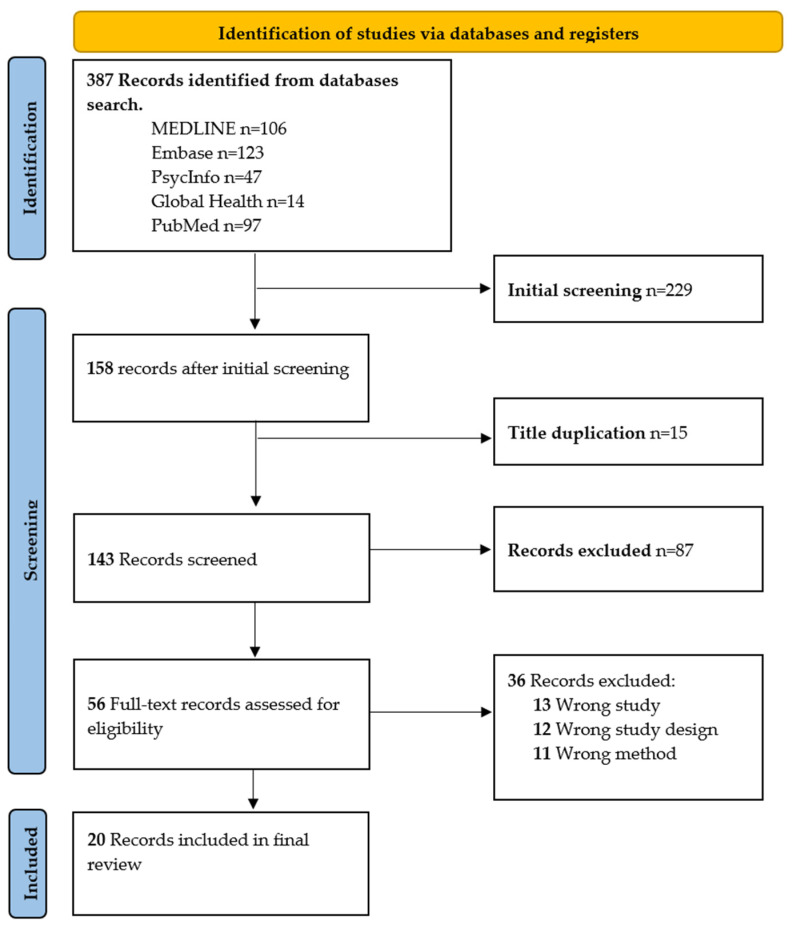
PRISMA Diagram.

**Figure 2 ijerph-20-01514-f002:**
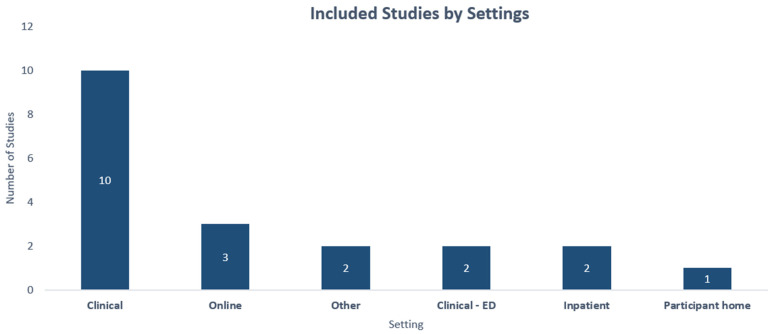
Studies by settings.

**Figure 3 ijerph-20-01514-f003:**
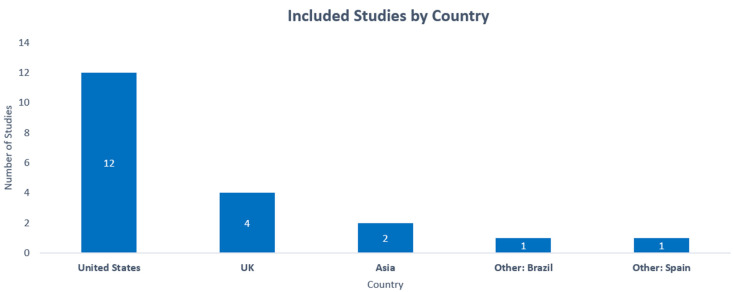
Studies by country.

**Table 1 ijerph-20-01514-t001:** Study Characteristics.

Study Title	Authors, Year	Country	Aim of Study	Study Category	Study Population	Sample Size	Setting	Methods Used	Model Evaluation Method	Main Findings & Results
Novel Use of Natural Language Processing (NLP) to Predict Suicidal Ideation and Psychiatric Symptoms in a Text-Based Mental Health Intervention in Madrid	Benjamin L. Cook et al., 2016 [5]	Other: Spain	The study aims to use NLP and machine learning to predict suicidal ideation and heightened psychiatric symptoms among adults recently discharged from psychiatric inpatient or emergency room settings in Madrid.	Suicide; Other	Adult	Adults 18+ discharged after self-harm from ED or short hospitalisation < 7 days (*n* = 1453)	Participant home	Applied NLP and ML (logistic regression) approach to text messages. The intervention was delivered by text messages, sent to participants. The text message included a link to a questionnaire and a mobile application to receive responses from the participants the mobile app used by participants to report things such as sleep, appetite, anger etc. STATA 14 for logistic regression prediction. NLP algorithm used 50% (half) of the sample for training.	The model was evaluated using positive predictive value (PPV), sensitivity, and specificity on the positive cases in the remaining 50% of the sample.	A total of 43% (*n* = 609) didn’t report suicide ideation. Participants who didn’t report suicide ideation slept 7.32 h compared with 6.86 of those who reported. Sleep quality is also higher for non-suicide. NLP using open-ended questions had a reasonably high predictive value for suicidal ideation. Data obtained from free-text responses to general questions about patients’ mental states could be used to predict suicidal ideation using NLP effectively. It is possible to use NLP based machine learning prediction methods to predict suicide risk as well as heightened psychiatric symptoms in free-text responses sent via mobile phone. The use of novel NLP methods may create low-cost and effective alternatives to traditional resource-heavy data monitoring systems.
Improving ascertainment of suicidal ideation and suicide attempt with natural language processing	Cosmin A. Bejan et al., 2022 [23]	United States	To demonstrate that NLP methods can be developed to identify suicide phenotypes in EHRs to enhance prevention efforts, predictive models, and precision medicine.	Suicide	3.4 million patients, 200 million clinical notes		Clinical	Google’s word2vec trained on 10 million clinical notes from EH extraction of seed keywords ‘suicide’ and ‘suicidal’.	A mixed-method of evaluation: Manual review and compared to diagnostic codes ICD10/11, PPV, Recall, F1 score, the area under the receiver operator curve (AUROC)	NLP demonstrating consistently excellent PPV (>95% for both outcomes). An ideal solution for ascertaining suicidal ideation and suicide attempt was provided by psychiatric forms when available in HER This NLP system can be applied to any unstructured clinical text common in EHRs and is feasible to apply at scale (~200 M notes here). This information retrieval approach would be portable to other health systems and has been used for the investigation of social determinants of health.
Identification of suicidal behaviour among psychiatrically hospitalized adolescents using natural language processing and machine learning of electronic health records	Nicholas J.Carson et al., 2019 [9]	United States	To develop and evaluate a machine learning algorithm using natural language processing of electronic health records to identify suicidal behavior among psychiatrically hospitalized adolescents.	Suicide	Adolescents 12–20 years	241 respondents	Inpatient	NLP analysis using Invenio software. Unstructured clinical notes were downloaded from the year preceding the index inpatient admission. Natural language processing identified phrases from the notes associated with the suicide attempt outcome random-forest machine-learning algorithm to develop a classification model.	Sensitivity, specificity, positive predictive value (PPV), negative predictive value (NPV), and accuracy	A moderate sensitivity and negative predictive value, a modest AUC, and accuracy below the most frequent class baseline.
Use of a Natural Language Processing-based Approach to Extract Suicide Ideation and Behaviour from Clinical Notes to Support Depression Research	Palmon N et al., 2021 [24]	United States	This study aimed to determine the feasibility of extracting SI from clinical notes.	Suicide	3.7 million patient notes		Clinical	Data were drawn from the OM1 Real World Data Cloud(OM1, Inc., Boston, MA, USA), derived from deterministically linked, de-identified, individual-level health care claims, EHR and other data from 2013 to the present day.		Extraction of SI is feasible. Future efforts should assess the reproducibility of this approach in other data sources and examine the feasibility of classifying SI as passive or active using data contained within the clinical notes.
Using natural language processing to extract self-harm and suicidality data from a clinical sample of patients with eating disorders: a retrospective cohort study	Charlotte Cliffe et al., 2021 [16]	UK	To determine risk factors for those diagnosed with eating disorders who report self-harm and suicidality.	Suicide; Other	Patients diagnosed with an eating disorder in South London and Maudsley	7188 patients	Clinical	NLP, STRATA software Analysed the data as an event notes in the EHRs, irre-spective whether they were created during an inpatient stay, during follow-up or a telephone appointment. The analysed cohort was extracted via the Clinical Record Interactive Search (CRIS) system and comprised of individuals who received an ICD-10 diagnosis of an ED (F50.0 and F50.9) within the 12-year observation period.	Manual annotations and calculating precision (PPV) and recall (sensitivity) statistics	Strong and near perfect agreement and when compared with manual annotations demonstrating the validity of the tool. This study also highlights the potential use of EHR databases to further suicidality and SH research by using NLP techniques. These tools could potentially have use with further development in risk prediction within ED services.
Integration and Validation of a Natural Language Processing Machine Learning Suicide Risk Prediction Model Based on Open-Ended Interview Language in the Emergency Department	Joshua Cohen et al., 2022 [25]	United States	To evaluate the performance of an NLP/ML suicide risk prediction model on newly collected language from the South-eastern United States using models previously tested on language collected in the Midwestern US To determine if the interview process to collect language for an NLP/ML model could be integrated into two EDs in the South-eastern United States, and (2) evaluate model performance on language from persons in a different geographic region than where the original model was developed	Suicide	ED patients 18–65 years	70 patients	Clinical—ED	37 suicidal and 33 non-suicidal patients from two EDs were interviewed to test a previously developed suicide risk prediction NLP/MLmodel. Model performance was evaluated with the area under the receiver operating characteristic curve (AUC) and Brier scores. Interview for data collection.	AUC and Brier scores AUC of 0.81 (95% CI = 0.71–0.91) and a Brier score of 0.23 when predicting suicidal risk on the 70 patient interviews collected in this study.	The language-based suicide risk model performed with good discrimination when identifying the language of suicidal patients from a different part of the USA and later period than when the model was originally developed and trained the study shows that integrating technology and procedures to collect language for a suicide risk prediction model into the ED workflow is feasible. A brief interview can be successfully implemented into two EDs and NLP/ML models can predict suicide risk from the patient language with good discrimination.
Natural Language Processing of social media as Screening for Suicide Risk	Glen Coppersmith et al., 2018 [26]	United States	The creation of an automated model for analysis and estimation of suicide risk from social media data. An examination of how this could be used to improve existing screening for suicide risk within the health care system. An exploration of the ethical and privacy concerns of creating a system for suicide risk screening not currently in care.	Suicide		418 users	Online	public self-stated data and using data donated through OurDataHelps.org Deep learning	10-fold cross-validation receiver operating characteristic (ROC)	These machine learning algorithms are of sufficiently high accuracy to be fruitfully used in an envisioned screening system, but the remaining parts of the system are not yet ready for implementation Although the design of an intervention system powered by algorithmic screening is technically possible, the cultural implications of implementation are far from settled Currently, this technology is only used for intervention for individuals who have opted in for the analysis and intervention, but the technology enables scalable screening for suicide risk, potentially identifying many people who are at risk preventively and prior to any engagement with a health care system.
Boamente: A Natural Language Processing-Based Digital Phenotyping Tool for Smart Monitoring of Suicidal Ideation	Evandro J S Diniz et al., 2022 [15]	Other: Brazil	To develop the Boamente tool, a solution that collects textual data from usersâ€™ smartphones and identifies the existence of suicidal ideation.				Online	NLP/Deep learning An android virtual keyboard can passively collect user texts and send them to a web service. We then developed a web platform composed of a service to receive texts from keyboard applications, a component with the DL model deployed, and an application for data visualization Twitter data, deep learning and evaluation 80 training and 20 testing	5-fold cross-validation	The proposed tool demonstrated an ability to identify suicidal ideation from user texts, which enabled it to be experimented with in studies with professionals and their patients. The performance evaluation results of the model selected to be deployed in the system (BERTimbau Large) were demonstrated to be promising. Therefore, the Boamente tool can be effective for identifying suicidal ideations from non-clinical texts, which enables it to be experimented with in studies with professionals and their patients.
Identifying Suicide Ideation and Suicidal Attempts in a Psychiatric Clinical Research Database using Natural Language Processing	Andrea C Fernandes et al., 2018 [27]	UK	To develop NLP approaches to identify and classify suicide ideation and attempts.	Suicide			Clinical	NLP approaches. A rule-based approach to classifying the presence of suicide ideation and a hybrid machine learning and rule-based approach to identify suicide attempts in a psychiatric clinical database. The Clinical Record Interactive Search (CRIS) system provides de-identified information sourced from South London and Maudsley (SLaM) NHS Trust Events and Correspondence document in CRIS EHR	Manually annotated gold standard set producing precision and recall statistics sensitivity of 87.8% and a precision of 91.7%	The good performance of the two classifiers in the evaluation study suggests they can be used to accurately detect mentions of suicide ideation and attempt within free-text documents in this psychiatric database. Two distinct NLP approaches are described to identify and classify suicide ideation and attempts, both of which performed well as indicated by high precision and recall statistics.
A Controlled Trial Using Natural language processing to Examine the Language of suicidal Adolescents in the emergency department	John P Pestian et al., 2016 [28]	United States	To design a prospective clinical trial to test the hypothesis that machine learning methods can discriminate between the conversation of suicidal and non-suicidal individuals.	Suicide	Children	60	Clinical—ED	NLP semi-supervised machine learning methods, the conversations of 30 suicidal adolescents and 30 matched controls were recorded and analysed. Questionnaire and interview for data gathering.		The results show that the machines accurately distinguished between suicidal and non-suicidal teenagers. The findings here support NLP as a strong adjunct to existing methods of determining a potentially suicidal individual.
Developing a Natural Language Processing tool to identify perinatal self-harm in electronic healthcare records	Karyn Ayre et al., 2021 [29]	UK	To create an NLP tool that can, with acceptable precision and recall, identify mentions of acts of perinatal self-harm within EHRs. (2) To use this tool to identify service-users who have self-harmed perinatally, based on their EHR.	Self-harm	Perinatal—18 years+		Clinical	NLP CRIS EHR	The evaluation was done against a manually coded reference standard. Precision and recall.	It is feasible to develop an NLP tool that identifies, with acceptable validity, mentions of peri-natal self-harm within EHRs, although with limitations regarding temporality.
Natural language processing of clinical mental health notes may add predictive value to existing suicide risk models	Maxwell Levis et al., 2020 [19]	United States	To evaluate whether natural language processing (NLP) of psychotherapy note text provides additional accuracy over and above currently used suicide prediction models.	Suicide	Veterans Health Administration VHA users diagnosed with PTSD	246 cases	Clinical	EHR stored in Data Warehouse, VA users newly diagnosed with PTSD least absolute shrinkage and selection operator (LASSO)	The area under the curve (AUC) and confidence interval (95%) statistics were calculated to determine the models’ predictive accuracy using the c-statistic.	NLP derived variables offered small but significant predictive improvement (AUC = 0.58) for patients with longer treatment duration. The small sample size limited predictive accuracy. Findings suggest leveraging NLP derived variables from psychotherapy notes offers an additional predictive value over and above the VHA’s state-of-the-art structured EMR-based suicide prediction model. Replication with a larger non-PTSD specific sample is required.
Use of natural language processing in electronic medical records to identify pregnant women with suicidal behaviour: towards a solution to the complex classification problem	Qui-Yue Zhong et al., 2019 [30]	United States	To develop algorithms to identify pregnant women with suicidal behaviour using information extracted from clinical notes by natural language processing (NLP) in electronic medical records.	Suicide	Clinical—Pregnant women	275,843	Clinical	Extracted diagnostic data from both structured codified data and unstructured clinical notes processed by NLP. We assessed the diagnostic validity of the algorithm against gold-standard labels obtained from manual chart reviews by psychiatrists and a trained researcher.	gold-standard validation AUC 0.83, PPV, NPV, and sensitivity for performance validation	Showed that mining unstructured clinical notes using NLP substantially improves the detection of suicidal behaviour. 9331 women screened positive for suicidal behaviour by either codified data (N = 196) or NLP (N = 9145). The addition of NLP resulted in an 11-fold increase in the number of pregnant women with suicidal behaviour.
Using natural language processing to improve suicide classification requires consideration of race	Nusrat Rahman et al., 2022 [10]	United States	To improve the accuracy of classification of deaths of undetermined intent and to examine racial differences in misclassification.	Suicide	10 years and older		Other	Natural language processing and statistical text analysis on restricted-access case narratives of suicides, homicides, and undetermined deaths in 37 states collected from the National Violent Death Reporting System (NVDRS).	ROC curves and area under curve AUC	Analysis reveals that identification of suicide among undetermined death cases with Black decedents can be greatly improved when modelled using race-specific death narratives; the rate is comparable with the prediction of suicide for White undetermined death cases there is strong evidence that NLP and automated coding methods could improve the detection of indications for suicide and might, in particular, help detection in settings where the death manner is prone to biases due to the decedent’s race.
Improving Prediction of Suicide and Accidental Death After Discharge from General Hospitals with Natural Language Processing	Thomas H McCoy Jr. et al., 2016 [31]	United States	To determine the extent to which incorporating natural language processing of narrative discharge notes improves stratification of risk for death by suicide after medical or surgical hospital discharge.	Suicide	845,417 discharges		Other	sociodemographic data, billing codes, and narrative hospital discharge notes for all patients from the hospital’s EHRs. NLP/statistical analysis	AUC 0.73	Automated tools to aid clinicians in evaluating these risks may assist in identifying high-risk individuals
Natural language processing and machine learning of electronic health records for prediction of first-time suicide attempts	Fuchiang R Tsui et al., 2021 [32]	United States	Aim to predict first-time suicide attempts using a large data-driven approach that applies natural language processing (NLP) and machine learning (ML) to unstructured (narrative) clinical notes and structured electronic health record (EHR) data.	Suicide	10–75 years	45,238	Clinical	Used both unstructured and structured data cTAKES NLP tool to process narrative notes.	ROC and AUC	Using both structured and unstructured EHR data demonstrated accurate and robust first-time suicide attempt prediction and has the potential to be deployed across various populations and clinical settings. Using recently developed NLP analyses of unstructured textual data in EHRs provided a significant boost to the overall accuracy of these ML models.
Identifying Suicidal Adolescents from Mental Health Records Using Natural Language Processing	Sumithra Velupillai et al., 2019 [33]	UK	To evaluate a simple lexicon and rule-based NLP approach to identify suicidal adolescents from a large EHR databases.	Suicide	Adolescents	200	Clinical	Develop a comprehensive manually annotated EHR reference standard and assessed NLP performance at both document and patient-level on data from 200 patients CRIS EHR.	PPV, recall, f1-score	Simple NLP approaches can be successfully used to identify patients who exhibit suicidal risk behaviour, and the proposed approach could be useful for other populations and settings. The approach shows promising results.
Screening pregnant women for suicidal behaviour in electronic medical records: diagnostic codes vs. clinical notes processed by natural language processing	Qui-Yue Zhong, 2018 [34]	United States	To examine the comparative performance of structured, diagnostic codes vs. natural language processing (NLP) of unstructured text for screening suicidal behavior among pregnant women in electronic medical records (EMRs).	Suicide	Women 10–64 years	5880	Clinical	NLP		The use of NLP substantially improves the sensitivity of screening suicidal behaviour in EMRs. However, the prevalence of confirmed suicidal behaviour was lower among women who did not have diagnostic codes for suicidal behaviour but screened positive by NLP. NLP should be used together with diagnostic codes for future EMR-based phenotyping studies for suicidal behaviour.
Detecting suicide risk using knowledge-aware natural language processing and counselling service data	Zhongzhi Xu et al., 2021 [35]	Asia	To develop a domain knowledge-aware risk assessment (KARA) model to improve our ability of suicide detection in online counselling systems.	Suicide		22,000 conversations	Online	De-identified dataset from an emotional support system established in Hong Kong, comprising 5682 Cantonese conversations between help-seekers and counsellors NLP approach.	Precision, recall and c-statistic (ROC-AUC)	The proposed model outperformed standard NLP models in various experiments, demonstrating good translational value and clinical relevance. The present study further confirmed that it is both possible and helpful to deploy an accurate, passive, and automatic suicide risk detection model for alerting counsellors to the presence of potential risk in a user’s content during the engagement process.
Comparisons of different classification algorithms while using text mining to screen psychiatric inpatients with suicidal behaviours	H Zhu et al., 2020 [36]	Asia	To compare the performance of methods based on text mining screen suicidal behaviours according to the chief complaint of the psychiatric inpatients	Suicide		3600	Inpatient	Electronic Medical Records of inpatients with mental disorders were collected. The text mining method was adopted to screen suicidal behaviours. The performances of different combinations of six algorithms and two-term weighting factors were compared under various training set sizes, which were assessed by precision, recall, F1-value and accuracy SVM, KNN, CART, Logistic Regression, RF, Adaboost	Precision, recall, F1-value and accuracy	Findings provided a practical way to automatically classify patients with or without suicidal behaviours before admission to the hospital, which potentially led to considerable savings in time and human resources for the identification of high-risk patients and suicide prevention.

## Data Availability

Not applicable.

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
