# Peer review of "Application of Natural Language Processing (NLP) in Detecting and Preventing Suicide Ideation: A Systematic Review"

_ijerph, 2023, doi:10.3390/ijerph20021514_

Round 1

Reviewer 1 Report

The paper is a well conceived survey of the studies about the use of NLP-based techniques in self-harm risk prediction and suicide prevention.

The text is clear and the review is, to my knowledge, well conducted and complete.

What strikes is that, on the one hand, the methodology of the papers' collection and selection is transparent, clearly illustrated and justified; on the other hand, the choice of focusing on NLP-based studies is not. What are the advantages, if any, of applying NPL-based methods to monitoring self-harm leanings and preventing suicide, compared to alternative text mining  methodologies? Is NPL simply the most widely used approach?      

In my opinion, the Authors should consider the available range of text mining methods and discuss the relevance on NLP techniques in this wider context.

In fact, an intense debate is going on about the application of (human) supervised or unsupervised text mining methods to sentiment and opinion analysis.

Usually, unsupervised methods are considered more advisable because they can process a large amount of data at relatively low managing cost. Nevertheless, when we deal with a matter as sensitive as detecting and preventing suicide attempts, efficacy and precision of the technique seem to be more important than managing cost.

Therefore, the choice of a method to focus on should be motivated according to its accuracy, compared to its potential competitors, and several studies assert that human supervised methods are more accurate than unsupervised ones, especially when an informal language is used (see sec. 3.5) or when different subgroups may be characterized by language specificities (see p. 28, lines 435-436).

Alternative human supervised tex mining methods have been successfully applied to sentiment and opinion analysis in health policy research: see, e.g., G.King - Y.Lu, Verbal Autopsy Methods with Multiple Causes of Death, Statistical Science, 23(1), 78-91.

Unfortunately, the Authors themselves acknowledge that assessing the efficacy of NLP applications on a quantitative scale is an unfeasible task (see sec. 4.1). Even more so, a comparison of potential efficacy between alternative text mining approach might help justify the attention devoted to NLP-based methods.

Minor remarks:

p.4: subsec. 2.1 and 2.2 have the same title. It is a bit misleading.

p.26, line 326: "investigation.". Please, remove "."

p.28, line 414: ".And applied" should be "and applied"

p.28, line 438: "White" is "white"

Reviewer 2 Report

The topic is interesting and also useful. There are some concerns regarding the review paper. The author is claiming that review uses mixed method research however the themes identified does not show any significant addition due to inclusion of mixed method studies. The intro and rational don’t discuss the context of nlp and machine learning.  The theme 3.4 don’t clearly show studies which clearly supports the theme. The whole result didn’t significantly point out that how nlp could be a best medium then other methods. Author in between says that demographics can’t helps to prevent suicide but at some places the findings have been discussed utilising demographics only. So overall the findings are contradictory in nature. The themes are useful and seems plausible but the discussion don’t support the findings much. 
the discussion too does not connect the themes well and don’t contextualise the findings in today’s context. 

Round 2

Reviewer 1 Report

Just a small remark: 

The last sentence of the paper argues that unsupervised methods require more time and resources to implement. I think the common opinion is the opposite: these methods, being mainly automatic tools, are less time-consuming and less costly.

Moreover, I could not find a clearly opposite statement in reference [63].
